# Impact of Antithyroperoxidase Antibodies (Anti-TPO) on Ovarian Reserve and Early Embryo Development in Assisted Reproductive Technology Cycles

**DOI:** 10.3390/ijms24054705

**Published:** 2023-02-28

**Authors:** Galina Kh. Safarian, Dariko A. Niauri, Igor Y. Kogan, Olesya N. Bespalova, Lyailya Kh. Dzhemlikhanova, Elena A. Lesik, Evgeniya M. Komarova, Inna O. Krikheli, Ksenia V. Obedkova, Nataliya N. Tkachenko, Yulia P. Milyutina, Aleksandr M. Gzgzyan, Yehuda Shoenfeld

**Affiliations:** 1D.O. Ott Research Institute of Obstetrics, Gynecology and Reproductive Medicine, St. Petersburg 199034, Russia; 2Medical Faculty, Saint Petersburg State University, St. Petersburg 199106, Russia; 3Zabludowicz Center for Autoimmune Diseases, Sheba Medical Center, Tel-Hashomer, Ramat Gan 5265601, Israel

**Keywords:** thyroid autoimmunity, infertility, follicular fluid, ovarian reserve, AT-TPO, embryo, IVF

## Abstract

Autoimmune thyroid disease (AITD) is one of the most common endocrinopathies and is more prevalent in women. It becomes evident that the circulating antithyroid antibodies that often follow AITD have effects on many tissues, including ovaries, and therefore that this common morbidity might have an impact on female fertility, the investigation of which is the aim of the present research. Ovarian reserve, ovarian response to stimulation and early embryo development in infertile patients with thyroid autoimmunity were assessed in 45 women with thyroid autoimmunity and 45 age-matched control patients undergoing infertility treatment. It was demonstrated that the presence of anti-thyroid peroxidase antibodies is associated with lower serum anti-Müllerian hormone levels and antral follicle count. Further investigation revealed the higher prevalence of sub-optimal response to ovarian stimulation in TAI-positive women, lower fertilization rate and lower number of high-quality embryos in this group of patients. The cut-off value for follicular fluid anti-thyroid peroxidase antibody affecting the above-mentioned parameters was determined to be 105.0 IU/mL, highlighting the necessity of closer monitoring in couples seeking infertility treatment with ART.

## 1. Introduction

Infertility of unknown origin remains one of the unresolved problems of modern reproductive medicine. Thus, in order to reduce the prevalence of idiopathic infertility, research projects aiming to identify reproductive and significant somatic pathology are actively performed. Some studies demonstrated an association between autoimmune thyroid disease (AITD) and gynecological pathology, leading to fertility decline [1,2,3]. There are also reports assessing the impact of anti-thyroid autoantibodies (ATA) on the efficacy of assisted reproductive technologies (ART) programs in euthyroid women with infertility [4,5,6]. The relevance of ATA in women undergoing in vitro fertilization (IVF) or intracytoplasmic sperm injection (ICSI) are controversial: some studies indicate a strict negative impact of the above-mentioned autoantibodies on the IVF/ICSI outcome [4,5,7], while others debate this assumption [8,9].

Currently, possible mechanisms by which anti-thyroid peroxidase (anti-TPO) antibodies affect the tissues of the reproductive system, namely, ovarian tissue, exist. It is known that ATA can cross the blood–ovarian barrier and are detected in the follicular fluid of women with infertility and thyroid autoimmunity [5,10]. Kelkar et al. observed a cross-reactivity between anti-*zona pellucida* antibodies obtained from sera of women with autoimmune oophoritis and mice thyroid tissue in experimental study [11]. Based on this finding, it was suggested that antithyroid antibodies could bind to zona pellucida, leading to altered early embryo development, i.e., fertilization and hatching, resulting in diminished implantation capability [10]. Interestingly, a direct negative impact of ATA on embryo quality was reported earlier by Lee at al., as the authors detected the above-mentioned antibodies on the surface of pre-implantation mice embryo [12].

Thus, the present study aimed to investigate the impact of autoimmune antibodies, namely, anti-TPO, detected in the sera and follicular fluids of women with infertility and autoimmune thyroiditis, on main ovarian reserve characteristics and embryological outcome of in vitro fertilization programs, contributing to better understanding of the mechanisms by which AITD is able to impair fertility in euthyroid women. Such understanding is crucial for successful pregnancy and live birth achievement.

## 2. Results

### 2.1. General and Hormonal Profile of the Patients Investigated

According to the results, patients of both groups were comparable by age, BMI and infertility duration (anti-TPO positive—4 (3; 5) years, anti-TPO negative group—4 (3; 6) years, *p* = 0.9). There was also no significant difference in hormonal parameters between the groups (Table 1). It has to be emphasized that no significant differences in serum TSH were noted between the groups, which is explained by L-thyroxine supplementation therapy at daily doses up to 75 mcg in 71.1% of patients with autoimmune thyroiditis. The mean serum values of antithyroperoxidase antibody in the anti-TPO+ group were equal to 481.8 (269.0; 723.2) IU/mL, in the anti-TPO− group—2.7 (1.0; 8.6) IU/mL.

Among the women included, 52.3% had primary infertility and 47.7% had secondary infertility. Main infertility reasons among couples investigated were: male, tubal, endocrine (anovulation) and endometriosis-associated infertility (Table 2). No significant differences in origin of infertility between the two groups were noted.

### 2.2. Correlation between Serum and Follicular Fluid AT-TPO Measured by ELISA

The quantitative measurement of AT-TPO by ELISA revealed the presence of the mentioned antibodies within the follicular fluid. The mean levels in the anti-TPO positive group were equal to 199.8 (85.9; 298.0) IU/mL. Subsequent Spearman’s correlation test revealed a strong positive association between serum and follicular fluid AT-TPO levels (Rs = 0.992, *p* = 0.00001). The result obtained is presented in Figure 1.

### 2.3. The Results of Ovarian Reserve Investigation in Relation to Follicular Fluid AT-TPO

Evaluation of the ovarian reserve among patients investigated revealed a reliable decrease in serum AMH levels among AT-TPO positive women relative to AT-TPO negatives. In the anti-TPO positive group, mean AMH values were 1.7 (1.4; 3.5) ng/mL, in anti-TPO negative group—3.6 (2.2; 6.3) ng/mL (*p* = 0.0007), while both groups were comparable by serum FSH. In addition, a significantly lower antral follicle count was noted among the antibody-positive group when compared to antibody-negative patients (8 (6; 10) vs. 11 (9; 17), respectively, *p* = 0.00007) (Figure 2 and Figure 3).

When analyzing an association between follicular fluid AT-TPO and ovarian reserve characteristics, a negative correlation was determined both with serum AMH (Rs = −0.272, *p* = 0.017) and antral follicle count before the start of controlled ovarian stimulation (Rs = −0.319, *p* = 0.0022).

### 2.4. Analysis of Controlled Ovarian Stimulation (COS) Efficiency

Controlled ovarian stimulation was achieved by administration of recombinant FSH (rFSH) preparations in 66 women (73.4%), and human menopausal gonadotropins (hMG)—in 11 patients (12.2%); combined rFSH and hMG preparations were prescribed in 4.4% of cases (13 patients). No significant differences were revealed between the groups in terms of preparations used for COS. Table 3 contains the main characteristics of controlled ovarian stimulation efficiency in IVF cycles among patients investigated.

Total dose, mean daily dose of gonadotropins and length of stimulation were comparable between the groups. However, the effective dose of gonadotropins was significantly higher in anti-TPO positives. In addition, the number of obtained oocytes was lower in the same group relative to anti-TPO negatives. The latter positively correlated with antral follicle count at the beginning of COS in all patients included (Rs = 0.731, *p* = 0.0081). Recombinant hCG preparation was used as final trigger in 73.4% of women, and GnRH agonist in 26.6% of patients. The rate of either final trigger administration did not significantly differ between the groups, indicating a comparable risk of ovarian hyperstimulation syndrome in both groups.

Nevertheless, our study did not reveal a significant correlation of total gonadotropin dose with follicular fluid levels of AT-TPO (Rs = 0.094, *p* = 0.261). However, the effective dose of gonadotropins was positively associated with follicular AT-TPO values (Rs = 0.315, *p* = 0.0026) among all patients included and was more prominent in the anti-TPO positive group (Rs = 0.366, *p* = 0.0022). In addition, in the same group the number of obtained oocytes was negatively associated with AT-TPO FF (Rs = −0.371, *p* = 0.0004).

### 2.5. Prognostic Model for Sub-Optimal Response to COS in ART Programs

Further exploration of ovarian response to COS among studied patients revealed significantly fewer normal responders (10–15 oocytes retrieved) among anti-TPO positive patients relative to negatives (24.4% vs. 71.1%; *p* = 0.0001). In contrast, sub-optimal response (≤9 oocytes retrieved) was noted in 26.7% of the anti-TPO negative group, while reaching 60.0 % in anti-TPO positives (*p* = 0.0001).

A logistic regression analysis was applied for determination of serum and follicular fluid AT-TPO levels associated with sub-optimal response to ovarian stimulation. According to the results obtained, the rate of sub-optimal response to COS increases in the presence of serum anti-TPO ≥ 305.0 IU/mL (OR 3.47 [CI 1.92–6.29], *p* < 0.01), as well as follicular fluid anti-TPO ≥ 105.0 IU/mL (OR 4.23 [CI 2.34–7.64], *p* < 0.01).

### 2.6. Evaluation of Early Embryologic Characteristics

Analysis of oocyte maturity revealed its significant decrease among women-carriers of anti-TPO relative to negative patients (5 (3; 9) vs. 9 (8; 11), *p* = 0.0004). The MII number was inversely associated with follicular fluid anti-TPO values (Rs = −0.538, *p* = 0.00006) among all patients included.

Table 4 contains the data regarding fertilization techniques used in the present study and demonstrates comparable results between IVF or ICSI.

Subsequent analysis of the mean 2PN zygote number revealed its significant decrease in anti-TPO positive women when compared to anti-TPO negatives (4 (2; 7) vs. 7 (5; 9), *p* = 0.001). Moreover, in the positive group, the number of 2PN zygotes was inversely correlated with follicular fluid values of anti-thyroid peroxidase antibodies (Rs = −0.286, *p* = 0.0168) (Figure 4), while the same trend was absent in negative patients.

Data on comparative analysis of daily embryological outcomes in the setting of IVF (IVF/ICSI) among both groups is presented in Table 5.

According to the results, the number of high-quality embryos on Day 3 and Day 5 of development in vitro were significantly lower among patient-carriers of anti-TPO in relation to negatives. The embryological assessment on Day 4 after fertilization determined a comparable number of morphologically high-quality embryos between the examined groups.

Further application of a follicular fluid anti-TPO threshold value of 105.0 IU/mL, associated with lower number of oocytes retrieved, revealed a significant decline in chances to achieve Day 3 (OR 0.15 [CI 0.07–0.35], *p* < 0.01) and Day 4 (OR 0.26 [CI 0.12–0.57], *p* < 0.01) high-quality embryos. At the same time, no reliable difference in the rate of Day 5 high-quality blastocysts depending on follicular fluid anti-TPO value was determined, probably due to the transfer of embryos on Day 3 and 4 of development, resulting in a small sample size.

## 3. Discussion

Overcoming infertility is a global medical and socially significant challenge; however, at present, in vitro fertilization program efficiency does not exceed 8.6–46.2%, and therefore, the investigation of factors that affect ART treatment outcome becomes highly relevant. In this context, an important place belongs to immunological markers, namely, circulating autoantibodies involved in the pathogenesis of both organ-specific and systemic autoimmune diseases directly affecting female fertility. Comprehension and prediction modeling of reproductive function alteration as well as determining its optimal management pattern are promising ways to increase the efficiency of IVF procedure.

Autoimmune thyroid disease (AITD) or autoimmune thyroiditis (AIT) are the most common organ-specific autoimmune disorders affecting 2–5% of the population in Western countries and among reproductive-aged women. It is found five to ten times more often in women than in men. The condition is characterized by the presence of serum antibodies directed against a membrane-associated hemoglycoprotein thyroperoxidase (anti-TPO) or a glycoprotein homodimer thyroglobulin (anti-Tg) [13].

The molecular mechanisms underlying the association between diminished fertility and the presence of thyroid antibodies are not fully understood. Currently, due to an active introduction of ART into clinical practice, it is evident that anti-thyroid antibodies are present in the follicular fluid of women with AITD and that their levels correlate with serum values [5,10]. Concentrations of these antibodies were about half of the serum concentrations after passing the blood–ovarian barrier. In women with ATA, decreases in the fertilization rate (63% vs. 72%, *p* = 0.05) and the proportion of high-quality embryos (25% vs. 48%, *p* = 0.05) were demonstrated as compared to negative controls. Interestingly, in ATA-positive women, pregnancy occurred only in those for whom the in vitro fertilization/intracytoplasmic sperm injection procedure (IVF/ICSI) was performed, while in the control group, pregnancy occurred both in the cases of classic IVF and ICSI. Generally, better outcome was observed in those with lower antibody concentrations, although the sample size was small [10]. Our results are in line with these studies, as we defined a reliable correlation between blood serum and follicular fluid levels of anti-thyroid peroxidase autoantibodies (Rs = 0.992, *p* = 0.00001) in infertile women affected by autoimmune thyroiditis. However, to our knowledge, previous studies did not report on anti-thyroid peroxidase antibody cut-off value in predictive modeling of ovarian response to stimulation, as well as early embryo development in ART cycles, among women with TAI. Recently, a strong correlation was observed between serum and follicular fluid thyroid hormone levels. Moreover, follicular fluid thyroxine (T4)/triiodothyronine (T3) ratio and serum TSH were positively associated with the number of oocytes retrieved, including mature ones, and embryos [14]. In this particular study, we did not investigate the follicular fluid levels of thyroid hormones; however, all patients included were in euthyroid state, highlighting the negative impact of autoimmune antibodies on ovarian tissue and ovarian response to stimulation within the framework of ART.

Currently, studies assessing the association of ovarian reserve in the presence of anti-thyroid autoantibodies are sparse. Hypotheses have been suggested to explain the possible connection between thyroid autoimmunity and ovarian reserve. According to one of them, ATA detection reflects a higher general susceptibility to autoimmunity, resulting in co-occurrence of thyroid autoimmunity with other autoimmune processes, such as, for example, early-phase autoimmune polyglandular syndrome I and II, resulting in hypogonadism and/or premature ovarian failure [11]. However, the most recent study conducted by Huang et al. on immunological changes within the follicles of women with autoimmune thyroiditis demonstrated a chemokine inflammatory cascade in an environment surrounding the maturing oocyte, namely, the increased levels of IFNγ-dependent chemokines: CXCL9, CXCL10, CXCL11 [15]. These molecules are known to be pro-inflammatory and carry out their chemotactic potential due to the directed migration of CXCR3 lymphocytes to the areas of inflammation, and mediate activated T-cell chemotaxis and their polarization. They are also characterized by a unique combination of inflammatory and angiostatic properties. The specified study showed the IFNγ dose-dependent stimulation of the CXCL9, CXCL10 and CXCL11 expression and secretion in the granulosa cell culture, as well as an increase in the CXCR3+ T-lymphocytes in the follicular fluid of women with AIT, indicating the immune imbalance within ovarian tissue. Thus, it turned out that the inflammatory process in the ovary is modulated by universal mechanisms typically occurring in organ-specific autoimmune diseases [16]. Considering the angiostatic properties of CXCL9, CXCL10 and CXCL11, as well as inflammatory infiltration with CXCR3+ T-lymphocytes, it is appropriate to assume the development of fibrotic changes within the ovary according to the general rules of inflammation, which in turn may affect the ovarian reserve. Indeed, some studies reported a decline in the main parameters of functional ovarian reserve among patients with AITD [8,17]. The present study, under otherwise equal conditions, revealed a reliable decrease in serum AMH and antral follicle count in the group of anti-TPO positive women relative to anti-TPO negative patients. However, opposite data exist of no association between anti-thyroid autoantibodies presence and ovarian reserve [18].

The success of infertility treatment is also known to be determined by the number of oocytes retrieved and requires nine to fifteen cells to achieve the best result [19]. The current study also assessed the association of gonadotropic ovarian stimulation outcome in the setting of ART programs and serum/follicular fluid anti-TPO among patients with AITD. As a result, the required dose of gonadotropins for obtaining one oocyte was reliably higher in the autoantibody-positive group relative to the antibody-negatives (*p* = 0.0004) and inversely correlated with the follicular fluid anti-thyroid peroxidase antibody values in the study group of patients (Rs = 0.366, *p* = 0.0022), reflecting lower ovarian sensitivity to gonadotropin stimulation. Although a number of studies deny the relation between antibody positivity and the number of oocytes retrieved in IVF cycles [4,5,20], our results indicate a significantly lower ovarian response in the study group of patients (*p* = 0.00008). In addition, the number of oocytes retrieved was negatively associated with follicular fluid anti-TPO levels (*p* = 0.0004). Within the framework of our study, a significant decrease in the rate normal responders to ovarian stimulation and increase in sub-optimal responders was determined among patients with AITD, regardless of the total dose administered. To the best of our knowledge, for the first time, the present study defined reliable follicular fluid anti-TPO values associated with lower ovarian response to stimulation in ART cycles and estimated their impact on the fertilization and embryo development. The same findings were reported in a recent retrospective study: significantly lower numbers of oocytes retrieved during IVF/ICSI treatment in infertile women with positive ATA were observed, despite higher serum AMH and estradiol levels in the same group of patients as compared to controls. The authors agree with the assumption that immune imbalance and abnormal thyroid function during controlled ovarian stimulation may affect follicle development in women with thyroid autoimmunity [21].

There is evidence that anti-thyroid autoantibodies may directly affect the *zona pellucida* of a maturing oocyte, resulting in its reduced quality and developmental potential [10]. Furthermore, data exist on decreased oocyte fertilization and top-quality embryo rates in TAI-positive women compared to those without TAI [5,22,23]. A meta-analysis, conducted in 2018, included four studies on ICSI outcomes only in women with or without autoimmune thyroiditis and revealed similar fertilization rates (combined OR 1.02, 95% CI [0.89–1.16], *p* = 0.09) in women of both groups. The authors assume that antithyroid antibody positivity may become a new independent indication for ICSI procedure regardless of the origin of infertility, as it may overcome the detrimental impact of the autoimmune condition on the embryo’s developmental potential [24]. Even though the present study did not report on fertilization rate, the numerical evaluation of 2PN zygotes obtained from women-carriers of anti-thyroid peroxidase autoantibodies revealed a reliable negative association with follicular fluid anti-TPO values, while no such pattern was noted in the negative group. Further analysis of the embryologic outcome, depending on threshold follicular antithyroperoxidase antibody level of 105.0 IU/mL, revealed a significantly reduced probability of obtaining morphologically high-quality Day 3 and Day 4 embryos (0.15 and 0.26, respectively). As a result, the number of top-quality blastocysts by Day 5 of development was reliably lower among women-carriers of AT-TPO (0.04). Our data are consistent with other studies and allow making an assumption about the direct influence of anti-thyroid antibodies on the oocyte and the quality of the developing embryo [4,10,22,25,26].

In addition, further studies with a larger sample size should be performed to confirm our findings and to elucidate the relationship between ovarian stimulation protocols, oocyte and embryo quality, and thyroid autoimmunity. In addition, determination of anti-TPO threshold values in prognostic modeling of ART clinical efficiency, i.e., clinical pregnancy and live birth rates, could be another promising way to increase the success rate of infertility treatment in women with autoimmune thyroid disease.

## 4. Materials and Methods

### 4.1. Patients

The study was approved by the ethics committee of “The Research Institute of Obstetrics, Gynecology and Reproductive medicine named after D. O. Ott” (protocol code 107 dated as of 15 March 2021) and performed at the Department of Assisted Reproductive Technologies. The recruitment period of participants was from September 2019 to March 2021. All participants gave informed written consent for participation.

In total, the results of 90 IVF or IVF/ICSI cycles were included. Inclusion criteria were the following: age 20–40 years old, infertility with the accepted indication for IVF or ICSI, random anti-Müllerian hormone (AMH) level ≥1 ng/mL, thyroid-stimulating hormone (TSH) level ≤2.5 mIU/L, serum anti-TPO levels ≥60 IU/mL, informed consent for participation. Patients with the body mass index ≤18 kg/m^2^ or ≥35 kg/m^2^, history of diabetes mellitus or any other endocrine disease (except autoimmune thyroiditis), history of thyroid or ovarian surgery, confirmed stage III–IV of endometriosis and karyotype abnormalities were excluded from the study.

Forty-five non-pregnant euthyroid women with infertility and verified autoimmune thyroiditis positive for anti-TPO were included in the study group. Forty-five women with infertility without any thyroid disorder and negative for thyroid autoimmunity (TAI) served as a comparison group. Routine pre-IVF evaluation at the center was performed on all women enrolled in the study.

### 4.2. Hormonal Evaluation

The evaluation included the measurement of serum follicle-stimulating hormone (FSH), luteinizing hormone (LH), anti-Müllerian hormone (AMH) and prolactin on the 2nd–3rd day of spontaneous menstrual cycle and was carried out using enzyme-linked immunosorbent assay (ELISA) and chemiluminescence immune assay. The quantitative determination of serum thyroid-stimulating hormone (TSH), free thyroxine (fT4) and anti-thyroid peroxidase antibody (anti-TPO) was implemented with ELISA and chemiluminescence immune assay irrespective of the day of menstrual cycle.

### 4.3. Sonographic Method

Pelvic sonogram was performed on all women recruited on the 2nd–3rd day of menstrual cycle before the beginning of ovarian stimulation. Biometric parameters of uterus, the structure and homogeneity of myometrium, the thickness and structure of endometrium, the size of ovaries and antral follicle count were estimated.

Later on, the frequency of ultrasound examination was individually determined based on the growth rate of follicles and endometrium in order to adjust gonadotropin doses used for ovarian stimulation, initiate daily GnRG antagonist injections and trigger ovulation.

The procedures of transvaginal needle aspiration and embryo transfer into uterus were performed under sonographic control. Thickness of endometrium of 0.8–1.2 cm and secretory transformation were considered optimal for embryo transfer. The clinical pregnancy was confirmed 3–4 weeks after embryo transfer by the presence of gestational sac in uterine cavity additionally to fetal heartbeat.

### 4.4. Ovarian Stimulation Protocol

A uniform “flexible” gonadotropin-releasing hormone antagonist protocol was applied for controlled ovarian stimulation starting on day 2–3 of menstrual cycle with daily injections of recombinant or urinary human gonadotropins. The starting dose of gonadotropins («Elonva», GmbH & Co.KG, Germany; «Gonal-F», Italy; «Puregon», The Netherlands; «Pergoveris», Italy; «Menopur», Germany) was estimated individually based on patient’s age, body mass index and ovarian reserve parameters.

The growth rate of follicles and endometrium was assessed during ultrasound examination. When three follicles reached a mean diameter of 13–14 mm, daily injections of GnRH antagonist were administered. When three of the leading follicles reached a mean diameter of more than 17 mm, a final trigger was introduced—human chorionic gonadotropin agent («Ovitrelle», Italy; «Pregnyl», The Netherlands) or GnRH agonist («Diphereline», France; «Decapeptyl» Germany). Follicle needle aspiration was performed using Cook Aspiration Unit under sonographic control.

Right after the aspiration of the follicular fluid into sterile, warmed-up test tubes, it was immediately passed to embryologist.

### 4.5. Fertilization and Early Embryo Development

Aspirated follicular fluid was evaluated for the presence of cumulus–oocyte complexes (COC) by embryologist under the stereomicroscope (×6 or 12 times) after its placement in Petri dish (diameter of 100 mm). The detected COC were transferred into prepared Petri dish 35 mm in diameter. After cumulus extraction, they were stored in incubator with constant parameters of temperature (37 °C) and media’s pH. The decision on fertilization method was defined by the ejaculate’s characteristics.

The fertilization was carried out using the IVF 3–5 h after oocyte pick-up if the sperm met criteria of normozoospermia (according to WHO criteria).

Intracytoplasmic sperm injection (ICSI) was used in following cases:–the presence of less than 5 million spermatozoa with progressive motility in 1 mL of ejaculate;–low sperm motility;–the presence of less than 4% spermatozoa with normal morphology (strict WHO criteria dated 2010);–high sperm agglutination rate—more than 80%;–low spermatozoa “swim up” rate;–high titer of antisperm antibodies (more than 60%);–low (less than 20%) or absent fertilization rate in previous IVF program.

Before ICSI, two hours after transvaginal needle aspiration, oocyte–cumulus complexes were denuded in a hyaluronidase solution, and after that, oocyte maturity was assessed under inverted microscope (×100). Oocyte with an absent polar body and preserved nucleus was a considered germinal vesicle in the first meiotic division. Metaphase I (MI) oocyte was characterized by both absent polar body and nucleus. GV and MI oocytes were considered immature. If a polar body was present in the perivitelline space, the oocyte was considered mature in the metaphase stage of second meiotic division (MII).

Fertilization was assessed 16–18 h later under inverted microscope (×10). Its efficiency was estimated by the presence of two pronuclei within the cytoplasm. If one or more than two pronuclei were seen, the fertilization was considered abnormal. Further embryo development took place in a CO_2_ incubator with constant temperature (37 °C), humidity and carbon dioxide concentration (5–6%).

Embryo quality was assessed on the 3rd (after 72 h), 4th (after 96 h) and 5th (120 h) days after fertilization based on morphological characteristics.

Day 3 embryo quality was assessed by blastomeres’ division rate, their symmetry and fragmentation [27]. Embryo quality on day 4 of development in vitro was conducted by the classification of Tao J. et al. (2002) according to blastomere compaction. Embryos grade 3 and 4 with blastomere compaction of more than 75% were considered of high quality [28]. The evaluation of embryos on day 5 of development in vitro was performed by Gardner D. K. system. In this classification, the degree of blastocyst size and its hatching has numeric expression from 1 to 6, and the assessment of inner cell mass and trophectoderm quality have literal expression (A–D in descending order). Blastocysts not less than 3BB were considered high-quality embryos [29].

### 4.6. Enzyme-Linked Immunosorbent Assay (ELISA) of Follicular Fluid

Follicular fluid was obtained on the day of oocyte pick-up. Samples not contaminated with blood were used for the analysis. After oocyte extraction, follicular fluid was centrifuged at 1000× *g* (*g*—relative centrifugal force unit, equals 980 m/s^2^) at room temperature for 15 min. Aliquoted samples of follicular fluid were then separated into sterile numerated Eppendorf tubes (0.5–0.6 mL) and stored at −80 °C until the assay. All samples were allowed to thaw at room temperature on the day of assay. A commercial ELISA kit was used for the follicular fluid investigation (Antithyroperoxidase (AT-TPO, E01T0531), Shanghai BlueGene Biotech Co., Ltd. (Shanghai, China).

After preparation, 100 μL of Standard or sample and Balance Solution were added to each well, mixed. A quantity of 50 μL of Conjugate was introduced, mixed, covered and incubated for 1 h at 37 °C. The solution was then aspirated, and the plate was washed 5 times. The next step included addition of 50 μL of Substrate and Substrate B, and incubation for 15–20 min at 37 °C. Finally, 50 μL of Stop Solution was applied and mixed, and immediate measurement of the results was conducted.

The obtained results on autoantibody levels were converted from ng/mL to IU/mL using standard conversion coefficient specific for anti-TPO antibody (0.6343 IU/mL).

### 4.7. Statistical Analysis

All statistical analyses were performed with STATISTICA 10.0 software. Kolmogorov–Smirnov test was used to evaluate the distribution of the parameters. Normally distributed measurement data were expressed as median (Me), lower quartile (LQ) and upper quartile (UQ). Data between the two groups were compared by the Mann–Whitney U test. Pearson chi-square test was used to compare the variables. The correlation between anti-TPO levels and ovarian reserve characteristics, as well as embryological outcomes, was determined using Spearman’s rank correlation as appropriate. Logistic regression analysis was used for determination of anti-TPO levels associated with lower embryological efficacy. For all tests, *p* value of <0.05 was considered statistically significant.

## 5. Conclusions

Despite the fact that the role of thyroid autoimmunity on natural conception and ART success rate remains debated, the current study allows to assume that antithyroid autoantibody positivity in euthyroid women with infertility, i.e., anti-thyroid peroxidase antibody, reflects a generalized immune response, as it exerts an effect on various steps of human reproduction starting from the ovarian tissue, composition of follicular fluid as a microenvironment for oocyte maturation, folliculogenesis and embryogenesis.

## Figures and Tables

**Figure 1 ijms-24-04705-f001:**
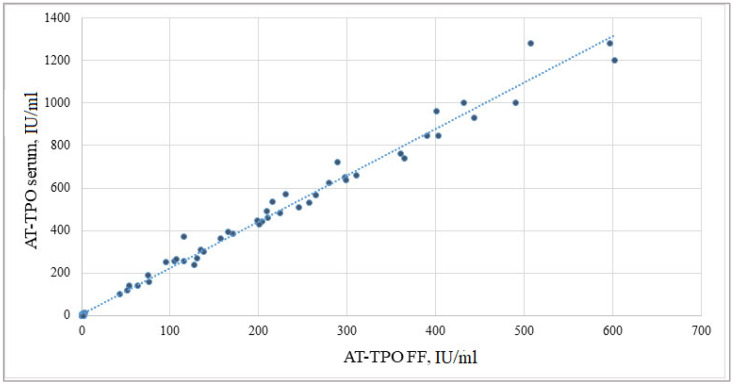
Correlation between serum and follicular fluid anti-thyroid peroxidase antibody levels. AT-TPO—anti-thyroid peroxidase antibody; FF—follicular fluid.

**Figure 2 ijms-24-04705-f002:**
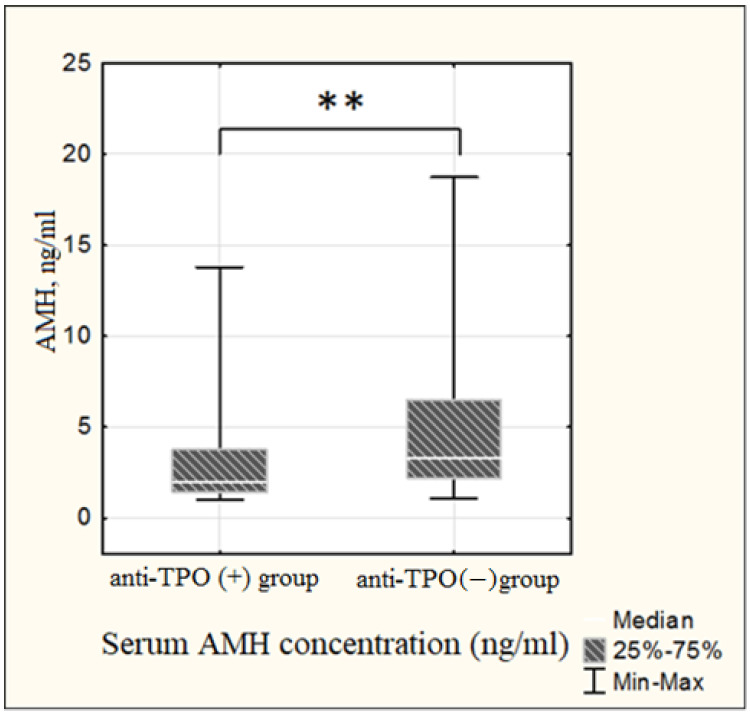
Comparative evaluation of serum AMH concentration in groups investigated. Mann–Whitney U-test, ** *p* < 0.01. AMH—anti-Müllerian hormone.

**Figure 3 ijms-24-04705-f003:**
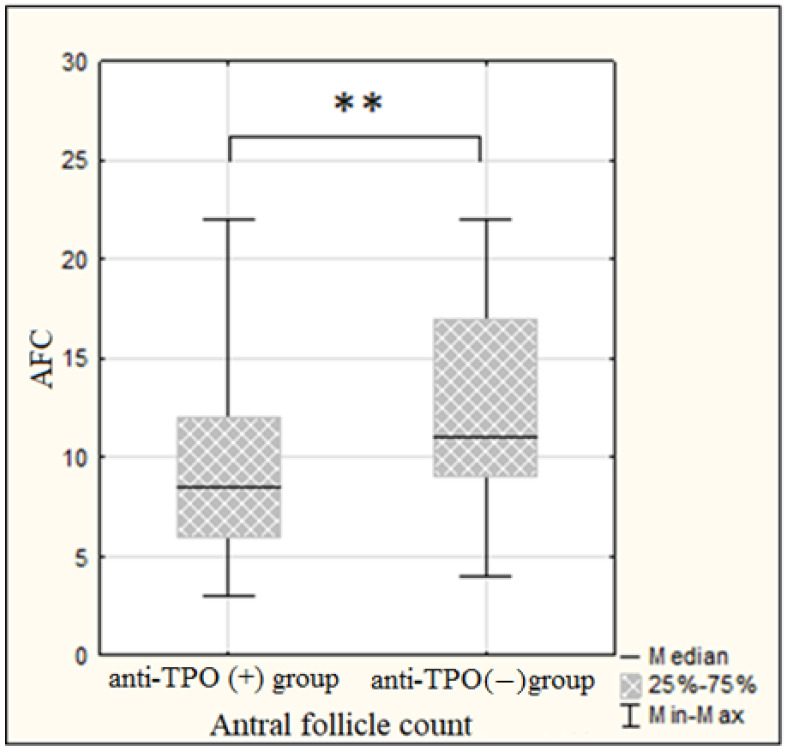
Comparative analysis of antral follicle count in the groups investigated. Mann–Whitney U-test, ** *p* < 0.01. AFC—antral follicle count.

**Figure 4 ijms-24-04705-f004:**
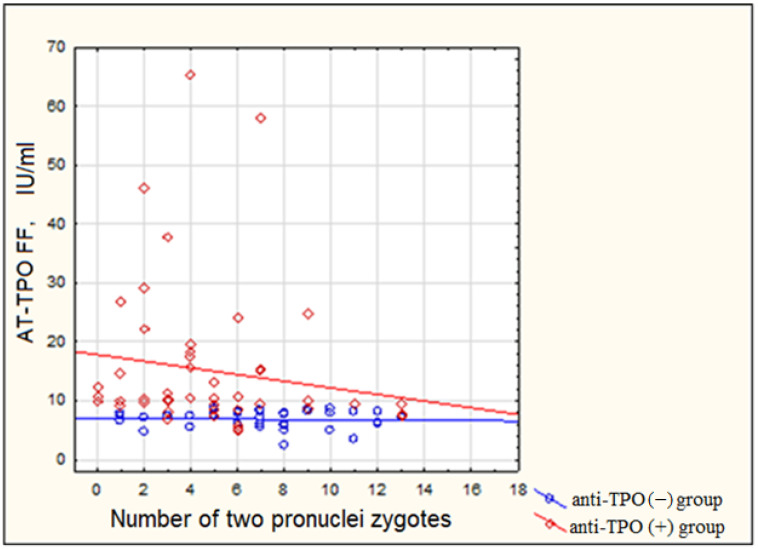
Association between the number of 2PN zygotes and follicular fluid AT-TPO values in both groups. AT-TPO—anti-thyroid peroxidase antibody; FF—follicular fluid.

**Table 1 ijms-24-04705-t001:** Comparative analysis of age-related, anthropometric and hormonal parameters within the groups.

Parameter	Anti-TPO+ Group (*n* = 45)	Anti-TPO− Group (*n* = 45)	*p*-Value
	Me (LQ; UQ)	Me (LQ; UQ)	
Age, years	35 (31; 37)	33 (31; 36)	0.3
Body mass index, kg/m^2^	24 (21; 28)	23 (21; 27)	0.5
Menarche, y.o.	13 (12; 14)	13 (12; 14)	0.2
FSH (mIU/mL)	7.8 (5.7; 9.5)	6.7 (5,3; 8.5)	0.1
LH (mIU/mL)	4.8 (3.5; 6.9)	5.7 (4.3; 7.4)	0.2
Prolactin (mIU/mL)	290.1 (139.0; 442.0)	339.5 (251.7; 430.0)	0.3
TSH (IU/L)	2.0 (1.6; 2.2)	1.9 (1.7; 2.3)	0.9
fT4 (pmol/L)	12.4 (11.3; 13.8)	12.0 (11.1; 14.2)	0.5

Mann–Whitney U-test. Me—median, LQ—lower quartile, UQ—upper quartile. FSH—follicle-stimulating hormone; LH—luteinizing hormone; TSH—thyroid-stimulating hormone; fT4—free thyroxine.

**Table 2 ijms-24-04705-t002:** Infertility causes within the groups.

		Anti-TPO+ Group (*n* = 45)		Anti-TPO− Group (*n* = 45)		*p*-Value
		*n*	%	*n*	%	
Infertility	Primary	20	44.4	27	60.0	0.1
	Secondary	25	58.1	18	40.0	0.1
Origin of infertility	Anovulation	2	4.4	4	8.9	0.4
	Tubal	15	33.3	14	31.1	0.8
	Male	22	48.9	25	55.6	0.5
	Endometriosis-associated	4	8.9	2	4.4	0.4
	Unknown origin	2	4.4	0	0.0	0.2

Pearson’s χ^2^ test, Yeats χ^2^ test, two-tailed Fisher’s exact test.

**Table 3 ijms-24-04705-t003:** Indicators of ovarian stimulation efficiency in groups investigated.

Parameter	Anti-TPO+ Group (*n* = 45)		Anti-TPO− Group (*n* = 45)		*p*-Value
	Me	LQ; UQ	Me	LQ; UQ	
Total dose of gonadotropins, IU	2025	1600; 2400	1800	1350; 2250	0.2
Mean daily dose of gonadotropins, IU	225	175; 300	200	150; 250	0.1
Effective dose of gonadotropins, IU	337.5	168.7; 533.3	166	110; 281.2	0.0004 ***
Stimulation length, days	9	8; 10	9	8; 10	0.9
Number of oocytes retrieved	7	4; 10	11	9; 13	0.00008 ***

Mann–Whitney U-test, *** *p* < 0.001. Me—median, LQ—lower quartile, UQ—upper quartile; IU—international unit.

**Table 4 ijms-24-04705-t004:** Comparative analysis of fertilization techniques used.

Parameter	Anti-TPO+ Group (*n* = 45)		Anti-TPO− Group (*n* = 45)		*p*-Value
	*n*	%	*n*	%	
IVF	21	46.7	19	42.2	0.23
ICSI	24	53.3	26	57.8	0.37

Pearson’s χ^2^ test. IVF—in vitro fertilization; ICSI—intracytoplasmic sperm injection.

**Table 5 ijms-24-04705-t005:** Comparative analysis of embryological outcome in groups investigated.

Parameter	Anti-TPO+ Group (*n* = 45)	anti-TPO− Group (*n* = 45)	*p*-Value
	Me (LQ; UQ)	Me (LQ; UQ)	
Day 3 high-quality embryos, *n*	3 (0; 4)	3 (2; 5)	0.02 *
Day 4 high-quality embryos, *n*	2 (0; 3)	2 (1; 4)	0.2
Day 5 high-quality embryos, *n*	0 (1; 2)	0 (2; 5)	0.04 *

Mann–Whitney U-test, * *p* < 0.05; Me—median, LQ—lower quartile, UQ—upper quartile; Day 3 embryo quality was assessed by blastomere’s division rate, their symmetry and fragmentation. Embryos grade 3 and 4 on day 4 of development with blastomere compaction of more than 75% were considered of high quality. Day 5 blastocysts not less than 3BB were considered high-quality embryos.

## Data Availability

The data that support the findings of this study are available on request from the corresponding author. The data are not publicly available due to privacy or ethical restrictions.

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
