# Peer review of "Impact of Antithyroperoxidase Antibodies (Anti-TPO) on Ovarian Reserve and Early Embryo Development in Assisted Reproductive Technology Cycles"

_ijms, 2023, doi:10.3390/ijms24054705_

Round 1

Reviewer 1 Report

This article “Impact of antithyroperoxidase antibodies (anti-TPO) on ovarian reserve and early embryo development in Assisted Reproductive Technologies cycles” aimed to investigate the impact of antithyroperoxidase antibodies (anti-TPO) on female fertility, these results are significative with sufficient experimental data and in line with readers' interests of IJMS. However, there are still some shortcomings that need to be improved or explained.

Comments:

Section 1: Abstract

Q1. In the last sentence, it is suggested to reinforce the guiding significance of this research for related fields.

Section 2: Introduction

Q2. The last paragraph could not well present the research significance, innovation, and necessity, which is suggested to be improved.

Section 3: Results

Q3. Since the authors aimed to investigate the impact of antithyroperoxidase antibodies (anti-TPO) on female fertility, selecting male volunteers seemed meaningless “Table 2”.

Q4. “Table 3. Indicators of ovarian stimulation efficiency in groups investigated”. n=45? Did male volunteers also provide these indicators? Similar confusion emerged in the other results.

Section 4: Discussion

Q5. Line 184-190, as the authors displayed, many researchers have proved that anti-thyroid antibodies are present in the follicular fluid of women with AITD and their levels correlate with serum values. Then whether the repeated testing was necessary? What is the relationship with these results?

Q6. Line 217-220, I considered that the differeces and reasons between these results and reported related data should be deeply analyzed and discussed.

Q7. In this part, the author should strengthen the summary of the results of this paper under the premise of referring to the existing reports.

Q8. The number of references is insufficient. It is recommended to supplement the relevant articles in the recent three years, which are generally no less than 30.

Author Response

Q1. Abstract. In the last sentence, it is suggested to reinforce the guiding significance of this research for related fields.

Response 1: Reinforecemt was added.

Q2. The last paragraph could not well present the research significance, innovation, and necessity, which is suggested to be improved.

Response 2: Information was added.

Q3. Since the authors aimed to investigate the impact of antithyroperoxidase antibodies (anti-TPO) on female fertility, selecting male volunteers seemed meaningless “Table 2”.

Response 3: Dear reviewer, you are absolutely right, the current study included only female patients. However, we have had couples with male origin of infertility (such patients were present in both groups). And, according to Table 4, the fertilization techniques used were comparable between the groups, allowing us to conclude the importance of antibodies, not the origin of infertility, on further early embryonic development.

Q4. “Table 3. Indicators of ovarian stimulation efficiency in groups investigated”. n=45? Did male volunteers also provide these indicators? Similar confusion emerged in the other results.

Response 4: Only female patients were included into the study. As previously mentioned, some couples were diagnosed with male origin of infertility. However, controlled ovarian stimulation was performed only in females, regardless the origin of infertility. In summary, no male patients were present in current study.

Q5. Line 184-190, as the authors displayed, many researchers have proved that anti-thyroid antibodies are present in the follicular fluid of women with AITD and their levels correlate with serum values. Then whether the repeated testing was necessary? What is the relationship with these results?

Response 5: Information was added.

Q6. Line 217-220, I considered that the differeces and reasons between these results and reported related data should be deeply analyzed and discussed.

Response 6: Dear reviewer, no lines are available to me. Also tried to deal this problem in the settings, but failed. I’ve added more results and discussions, but not sure if I did it the lines you’ve mentioned. May I kindly ask you to specify the exact place by indicating the sentences from the text? Appreciate it.

Q7. In this part, the author should strengthen the summary of the results of this paper under the premise of referring to the existing reports.

Response 7: Information was added.

Q8. The number of references is insufficient. It is recommended to supplement the relevant articles in the recent three years, which are generally no less than 30.

Response 8: Information was added.

Reviewer 2 Report

The manuscript presents an interesting and well-done study.

Please describe with more detail the legends of the figures including the acronym meaning. 

At the discussion please add the comparison results with similar studies. Some of the few studies on the subject are not cited. Look in Pubmed for instance.

Author Response

Point 1: Please describe with more detail the legends of the figures including the acronym meaning. 

Response 1: Details were added

Point 2: At the discussion please add the comparison results with similar studies. Some of the few studies on the subject are not cited. Look in Pubmed for instance.

Response 2: Information was added

Reviewer 3 Report

The authors investigated the role of anti-TPO on fertility response. The paper idea is novel and interesting. The article is well written.

In mdpi journals, methods used to be the second section.

Recommend adding abbreviations in footer of tables, and better specifying the group names in a more clear way, e.g positive anti-TPO versus negative anti-TPO.

Table 5 lacks header. Please describe the definition of high quality in the footer.

There is a typo in Line 165.

Suggest running multivariate regression analysis to predict the outcome.

Limitation section in the discussion is suggested.

Author Response

Point 1: In mdpi journals, methods used to be the second section.

Response 1: Dear reviewer, the article’s template was downloaded from the IJMS official website. In this template material and methods are mentioned in section 4.  

Point 2: Recommend adding abbreviations in footer of tables, and better specifying the group names in a more clear way, e.g positive anti-TPO versus negative anti-TPO.

Response 2: Abbreviations were added, group names were changed.

Point 3: Table 5 lacks header. Please describe the definition of high quality in the footer.

Response 3: Header for table 5 was added. Description of high quality embryos was described in the footer, more detailed description is in section 4.5. 

Point 4: There is a typo in Line 165.

Response 4: Dear reviewer, I’ve tried downloading the document on several PCs, however no lines are available to me. Also tried to deal this problem in the settings, but failed. I have fixed several typos, but not sure if I did the one you’ve mentioned. May I kindly ask you to specify the exact place by indicating the sentences from the text? Appreciate it.

Point 5: Suggest running multivariate regression analysis to predict the outcome.

Response 5: Unfortunately, it is not very clear which particular outcome is recommended to be predicted by multivariate regression. The marker indicated in the manuscript was used for logistic regression analysis in order to predict the sub-optimal response COS. We also used a multiple linear regression in order to predict the 2PN zygote number, unfortunately, despite the presence of significant correlations in the AT-TPO + group, a reliable regression model for the entire patient sample cannot be reached, and the levels of significance for serum and follicular fluid anti-TPO are 0.08 and 0.06, respectively (i.e. p> 0.05). Also, we consider that the application of AMH is not mandatory due to the well-known fact on the relationship between it’s level and the number of oocytes retrieved and, consequently, number of 2PN zygotes.

Point 6: Limitation section in the discussion is suggested.

Response 6: Limitation and further research strategies were added to the discussion.

Reviewer 4 Report

This study aimed to ascertain whether antithyroid antibodies (ATA), developed during autoimmune thyroid disease (AITD), could affect female fertility. In this regard, 45 infertile women with AITD were tested for ovarian reserve, ovarian response to stimulation and early embryo development in comparison with 45 age-matched patients undergoing infertility treatment. Antithyroperoxidase antibodies (anti-TPO, AT-TPO) were found to be associated with a decrease of both serum levels of anti-Müllerian hormone (AMH) and antral follicle count. Moreover, in TAI-positive women, there were sub-optimal responses to ovarian stimulation, lower fertilisation rate and reduced number of high quality embryos. A cut-off for anti-TPO, in the follicular fluid, influencing the above parameters was established in 105.0 IU/ml.

Introduction underlines literature data focusing on how ATA are able to affect reproductive structures, mostly the ovarian tissue where such antibodies bind to zona pellucida, thus leading to  negative consequences including impaired early embryo development and implantation capability.

Materials and Methods describe a) Selection of patients, i.e. women with infertility and autoimmune thyroiditis positive for AT-TPO and women with infertility without thyroid disorders b)

 Hormonal evaluation, i.e. determination of a series of hormones (FSH, LH, AMH, LTH, fT4, TSH) and AT-TPO c) Pelvic sonographic procedures allowing to also verify structure of endomethrium, size of ovaries, antral follicles and embryo transfer into uterus d) Ovarian stimulation protocol including human gonadotropin employment followed by GnRH antagonist administration and follicular fluid  needle aspiration e) Fertilization and early embryo development following cumulus-oocyte complexes extraction from follicular fluid, intracytoplasmic sperm injection and embryo transfer in suitable incubator  f) ELISA of follicular fluid to detect autoantibodies g) Employed statistical tests (such as the Mann U Whitney, Pearson chi-square and Spearman ones).

Results report a) general and hormonal profiles of the studied patients b) correlation between serum and follicular AT-TPO c) ovarian reserve with reference to follicular fluid AT-TPO d) ovarian stimulation efficiency (reduced in AT-TPO positive patients and sub-optimal in AT-TPO negative ones) e) evaluation of early embryologic characteristics in relation to the used fertilization techniques (with declined chance to achieve Day 3 high-quality embryos at the follicular fluid AT-TPO threshold value of 105.0 IU/ml).

Discussion correctly analyses the obtained results that appear to corroborate the hypothesis that the presence of anti-thyroid antibodies in the follicular fluid of women with autoimmune thyroid disease represents a significant factor of impaired fertility. In this regard, several studies (mentioned by Authors) evoke pathogenetic mechanisms involving a chemokine inflammatory cascade with increased levels of IFN-gamma-dependent chemokines characterized by pro-inflammatory and angiostatic properties. Moreover, the present study could contribute to assess gonadotropic ovarian stimulation outcome in the setting of ART programs, mostly in having defined the (above) reliable follicular fluid AT-TPO value that accompanies a reduced ovarian response to stimulation. It was concluded that, in euthyroid women with infertility, the positivity of antithyroid peroxidase antibodies negatively affects various steps of human reproduction in reflecting a generalized immune response.

Overall, this study contributes to a better understanding of the mechanisms by which AITD is able to impair fertility in euthyroid women. The same study is well planned and performed through accurate methodological procedures. Results, discussion and conclusion are concordant with the obtained experimental evidences. The presented three figures and five tables have been adequately prepared, resulting clear and comprehensive. The manuscript does not show censurable incongruities as well as lexicon, grammar, “English style” and sentence fluency are concerned; only sporadic refinements might be done in some sentences. References are adequate.

Author Response

Dear Reviewer,        

The whole list of authors would like to express their gratitude for your kind words and high rating of the research project. Appreciate it. Be safe.
